NATURE COMMUNICATIONS logo

# Mutations in *SMARCB1* and in other Coffin–Siris syndrome genes lead to various brain midline defects

Alina Filatova [1], Linda K. Rey[2], Marion B. Lechler[1], Jörg Schaper[3], Maja Hempel[4], Renata Posmyk[5], Krzysztof Szczaluba[6], Gijs W.E. Santen[7], Dagmar Wieczorek[2] & Ulrike A. Nuber [1]

Mutations in genes encoding components of BAF (BRG1/BRM-associated factor) chromatin remodeling complexes cause neurodevelopmental disorders and tumors. The mechanisms leading to the development of these two disease entities alone or in combination remain unclear. We generated mice with a heterozygous nervous system-specific partial loss-of-function mutation in a BAF core component gene, *Smarcb1*. These *Smarcb1* mutant mice show various brain midline abnormalities that are also found in individuals with Coffin–Siris syndrome (CSS) caused by *SMARCB1*, *SMARCE1*, and *ARID1B* mutations and in *SMARCB1*-related intellectual disability (ID) with choroid plexus hyperplasia (CPH). Analyses of the *Smarcb1* mutant animals indicate that one prominent midline abnormality, corpus callosum agenesis, is due to midline glia aberrations. Our results establish a novel role of *Smarcb1* in the development of the brain midline and have important clinical implications for BAF complex-related ID/neurodevelopmental disorders.

[1] Stem Cell and Developmental Biology, Technical University Darmstadt, Darmstadt 64287, Germany. [2] Institute of Human Genetics, Medical Faculty, Heinrich Heine University, Düsseldorf 40225, Germany. [3] Department of Diagnostic and Interventional Radiology, Medical Faculty, Heinrich Heine University, Düsseldorf 40225, Germany. [4] Institute of Human Genetics, University Medical Center Hamburg-Eppendorf, Hamburg 20246, Germany. [5] Podlaskie Medical Centre "GENETICS" Bialystok and Department of Perinatology and Obstetrics, Medical University of Bialystok, Bialystok 15-276, Poland. [6] Department of Medical Genetics, Medical University Warsaw, Warsaw 02-106, Poland. [7] Department of Clinical Genetics, Leiden University Medical Center, Leiden 2333 ZA, Netherlands. Correspondence and requests for materials should be addressed to U.A.N. (email: nuber@bio.tu-darmstadt.de)

In the course of mammalian development, a multitude of cell type specification, maintenance, and differentiation events takes place, governed by specific gene expression patterns. Chromatin remodeling complexes play an essential role in controlling the establishment and maintenance of gene activities. The relevance of these complexes is reinforced by the consequences of aberrations in their components.

Mutations in genes encoding components of the ATP-dependent chromatin remodeling BAF complex (mammalian SWI/SNF complex) are found in two distinct disease types: intellectual disability (ID)/neurodevelopmental disorders and tumors. They belong to the most frequently mutated genes in human ID, and close to 20% of all human cancers harbor mutations in these genes[1–4].

One of the disorders caused by mutations in BAF complex component genes is Coffin–Siris syndrome (CSS), a congenital malformation syndrome characterized by severe developmental delay affecting motor and intellectual functions, growth impairment, hypotonia, a distinct facial appearance with coarse features, feeding difficulties in infancy, and hypoplastic to absent fifth distal phalanges, fingernails and toenails[5]. Heterozygous germline mutations in six genes encoding BAF complex components have been found in CSS: *ARID1A*, *ARID1B*, *DPF2*, *SMARCA4*, *SMARCE1*, and *SMARCB1* (also referred to as BAF47, INI1, SNF5)[6–9]. Several of these genes are also associated with other ID/neurodevelopmental disorders, and owing to overlapping clinical features, their differentiation from CSS can be difficult[10]. A particular heterozygous *SMARCB1* mutation has been described in individuals with severe ID and choroid plexus hyperplasia (CPH) and we refer to this condition as *SMARCB1*-related ID-CPH[11]. It remains unresolved how alterations in BAF complex genes lead to the specific developmental brain defects seen in affected individuals. Heterozygous germline mutations in these genes also predispose to diverse tumors, including brain tumors[2,12], and it is unclear how such distinct disease entities can be caused by mutations in identical genes[3].

We generated mice (*Smarcb1*[+/inv] *NesCre*[+/−]) with a heterozygous reversible *Smarcb1* disruption in neural stem/progenitor cells. To our knowledge, this is the first mouse model that recapitulates brain defects seen in *SMARCB1*-related CSS and ID-CPH and is based on a monoallelic *Smarcb1* mutation. *Smarcb1*[+/inv] *NesCre*[+/−] mice show brain midline abnormalities and we identified similar alterations in CSS individuals. The spectrum of brain midline abnormalities in these individuals is much broader than described for CSS so far and includes reduction or absence of forebrain commissures, absence or hypoplasia of the septum pellucidum, choroid plexus (CP) hyperplasia, and cerebellar vermis hypoplasia. A prominent aberration in *Smarcb1* mutant animals, agenesis of the corpus callosum (CC), can be explained by defects in midline radial glial cells (RGCs). Interestingly, *Smarcb1* transcript levels are reduced by about 30% in embryonic brain tissue of these mutant mice, whereas there is no reduction in *Smarcb1* transcript levels in heterozygous germline *Smarcb1* knockout animals (*Smarcb1*[+/−]). *Smarcb1*[+/−] mice do not show any brain abnormalities but are prone to develop malignant tumors, presumably upon *loss-of-function* of the second *Smarcb1* allele[13]. Based on our results and on the review of individuals with *SMARCB1* mutations in the literature, we suggest that the development of ID/neurodevelopmental disorders or malignant tumors could depend on a transcriptional compensation by the second intact *SMARCB1* allele.

## Results

### Neurodevelopmental phenotypes in *Smarcb1*[+/inv] *NesCre*[+/−] mice.

We generated mice with a heterozygous *Smarcb1* disruption in the nervous system (*Smarcb1*[+/inv] *NesCre*[+/−]) by crossing *Smarcb1*[inv/inv] animals, in which exon 1 is flanked by loxP sites of opposite orientation[14] (originally termed *Snf5*[inv/inv]), with *NesCre*[+/−] animals expressing *Cre* recombinase under the control of the rat nestin promoter and nervous system-specific enhancer[15] (Fig. 1). *Cre* activity in these mice starts in nervous system tissue at embryonic day 10.5 (E10.5)[16]. The majority of targeted cells are therefore RGCs. *Cre*-mediated recombination leads to an inversion of the floxed sequence in *Smarcb1*[+/inv] *NesCre*[+/−] animals, resulting in an inactivation of this allele[14] (Fig. 1f). The inversion is reversible and continues forth and back as long as *Cre* is expressed, theoretically leading to about 50% of the *Nestin*-expressing cells possessing an active and 50% possessing an inactive *Smarcb1* allele, in addition to the second wild-type allele. Breedings with male or female *NesCre*[+/−] mice resulted in *Smarcb1*[+/inv] offspring according to Mendelian ratios (~50% *Smarcb1*[+/inv] *NesCre*[+/−] and ~50% *Smarcb1*[+/inv] *NesCre*[−/−]). *Smarcb1*[+/inv] *NesCre*[−/−] animals are named controls since they lack the *NesCre* transgene and do not show any obvious differences to wild-type mice.

*Smarcb1*[+/inv] *NesCre*[+/−] mice, referred to as mutant animals, were smaller (Fig. 1a) and had a lower body weight in comparison to littermate controls (Supplementary Fig. 1a). Very striking was their overall smaller brain with cerebral hemispheres widely separated from each other (Fig. 1b). The microcephaly was already apparent in utero: at E14.5, the brain weight of mutants was reduced by 41% in comparison to littermate controls, whereas there was no difference in body weight (Supplementary Fig. 1b, c). Brain phenotypes ranged from moderate ones with cerebral hemispheres rostrally, but not caudally connected, to extreme ones with rostral and caudal cerebral hemispheres fully separated, being only connected by thin dorsal membrane-like structures (Supplementary Fig. 1d). About 80% of mutant animals died before postnatal week 9, and such animals typically showed more severe brain phenotypes. The remaining ones survived for >1 year (Supplementary Fig. 1e) and were fertile (Supplementary Fig. 2a). To exclude that the brain phenotypes are caused by the *NesCre* allele, we performed additional breeding with mice of different genotypes (Supplementary Fig. 2a). Again, *Smarcb1*[+/inv] *NesCre*[+/−] animals were affected but not *Smarcb1*[+/+] *NesCre*[+/−] as well as other genotypes. In the course of our whole study, we once obtained a stillborn animal with a *Smarcb1*[inv/inv] *NesCre*[+/−] genotype. Embryonic brains of such mice with two mutant *Smarcb1* alleles were of severely reduced size (Supplementary Fig. 2b).

The cerebellum of 22 investigated postnatal mutants was hypoplastic and showed a reduced foliation (Fig. 1b, d). In addition, 19 of the 22 animals presented with particular cerebellar midline defects—macroscopically they showed a midline fusion of the cerebellar hemispheres without any longitudinal fissures (arrowheads in Fig. 1d). In the majority of cases, no vermis was seen at the cerebellar surface. In two cases, the cerebellar hemispheres were widely separated with only a superior tissue bridge present in between (arrow in Fig. 1d). Analyses of tissue sections confirmed a midline fusion of the cerebellar hemispheres and revealed a midline structure resembling a smaller vermis (Fig. 1e); however, no fissure separated this structure from the hemispheres as it is normally the case. We interpret this phenotype as a fusion/continuity of the vermis with the cerebellar hemispheres.

To further characterize the forebrain midline defects, we analyzed postnatal brain tissue and sections thereof (summarized in Fig. 1c). All 16 investigated mutants lacked the hippocampal commissure, 9 lacked the caudal CC, 6 showed a complete CC agenesis, and an intact CC was present in 1 case (Figs. 1c and 2b–d). The anterior commissure was absent in 6 of 16 cases (Figs. 1c and 2c, d). Above the third ventricle, the medial cortex

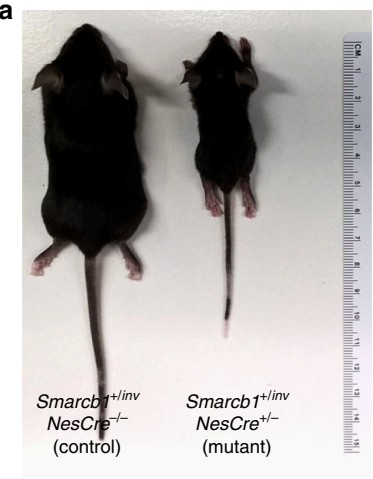

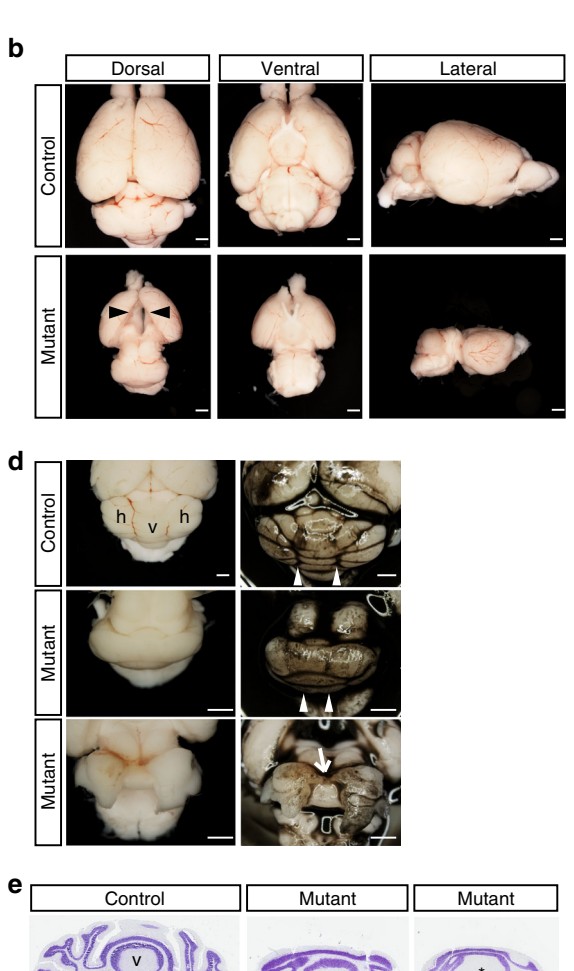

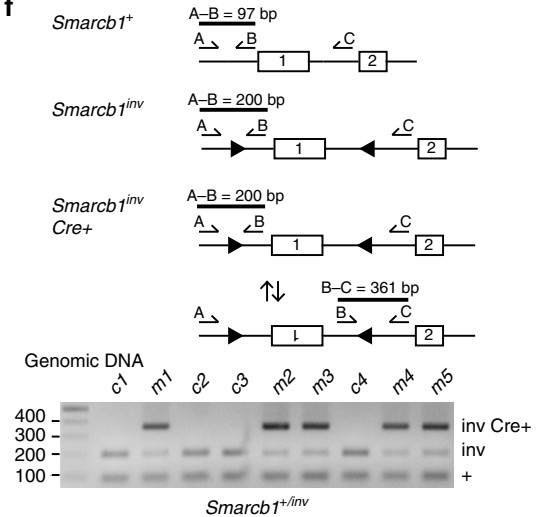

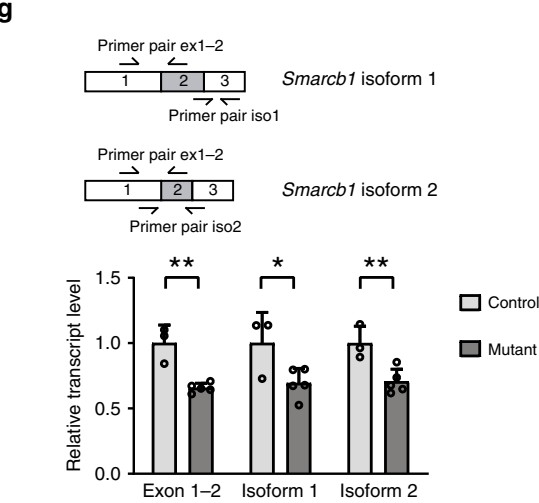

| | |
|---|---|
| Olfactory bulbs present | 22/22 |
| Cerebral hemispheres | |
| Hemispheres partially fused (rostrally) | 2/22 |
| Hemispheres not fused | 20/22 |
| Cerebral commissures | |
| Corpus callosum (CC) | |
| Partial CC agenesis | 9/16 |
| Complete CC agenesis | 6/16 |
| Anterior commissure (AC) | |
| AC present | 10/16 |
| AC absent | 6/16 |
| Hippocampal commissure (HC) | |
| HC agenesis | 16/16 |
| Cerebellum | |
| Cerebellar hypoplasia | 22/22 |
| Reduced lobulation | 22/22 |
| Vermis hypoplasia | 21/22 |
| Fusion of cerebellar hemispheres | 19/22 |
| Complete fusion | 16/22 |
| Partial fusion | 3/22 |

edges were located far apart from each other and only connected by thin membrane-like structures (compare Fig. 2a to Fig. 2b–d). In severe cases, this separation continued in a rostral direction (Fig. 2c, d). Furthermore, a midline cleft separated the left and right half of the diencephalon (arrows in Fig. 2b–d).

To better understand the consequences of the monoallelic reversible *Smarcb1* exon 1 inversion, we analyzed *Smarcb1* genomic DNA and transcript levels in E14.5 forebrain tissue. These experiments confirmed a *Cre*-mediated inversion of *Smarcb1* exon 1 (Fig. 1f) and revealed that *Smarcb1* transcript

**Fig. 1** Phenotypic and genetic analyses of mutant and control mice. **a** Control and mutant animal (age: 3 weeks). **b** Brains of 3-week-old mutant mice are smaller and their cerebral hemispheres are separated by a midline gap (arrowheads). **c** Summary of brain phenotypes in mutant animals. **d** Cerebelli of 3–4-week-old control and mutant animals; right panel: brains were ink-stained for the visualization of fissures; h cerebellar hemisphere, v cerebellar vermis. Note the smaller size and reduced foliation of mutant cerebelli. Upper panel: control brain with longitudinal (anterior-posterior) fissures (arrowheads) separating the vermis from the two lateral hemispheres. Middle panel: mutant cerebellum with lack of longitudinal fissures (arrowheads) and fusion of the two lateral hemispheres (the majority of cases). Lower panel: mutant cerebellum with separated hemispheres that are only rostrally connected (arrows). **e** Nissl-stained coronal sections of control and mutant cerebelli (3–4 weeks of age). Note the midline fusion of lateral hemisphere lobules in the mutants. v cerebellar vermis; asterisk (*) vermis-like midline structure. Scale bars in **b**, **d**, **e**: 1 mm. **f** Inversion of *Smarcb1* exon 1 in genomic DNA from E14.5 mutant and control forebrain tissue detected by polymerase chain reaction (PCR). Three primer pairs are used in one reaction, leading to amplification of a 97-bp product in case of the wild-type allele (*Smarcb1*+), of a 200-bp product in case of the active, non-inverted, floxed allele (*Smarcb1*inv) due to loxP site and extra DNA sequences, and of a 361-bp product in case of the inactive inverted allele (*Smarcb1*inv Cre+). Gel electrophoresis of the PCR products shows the presence of inverted and non-inverted alleles in all mutant samples (m1–m5). **g** *Smarcb1* transcript levels in E14.5 control (c2–c4) and mutant (m1–m5) forebrain tissue determined by quantitative reverse transcriptase-PCR. Primers binding to exons 1 and 2 and primers that specifically amplify one of the two *Smarcb1* isoforms differing in exon 2 length were applied. Significantly reduced levels of all three cDNA PCR products were found in mutant (five animals) compared to control (three animals) samples. All bars display the mean with standard deviations. *$p \leq 0.05$, **$p \leq 0.01$ (unpaired two-tailed Student's *t* test). Source data are provided as a Source Data file

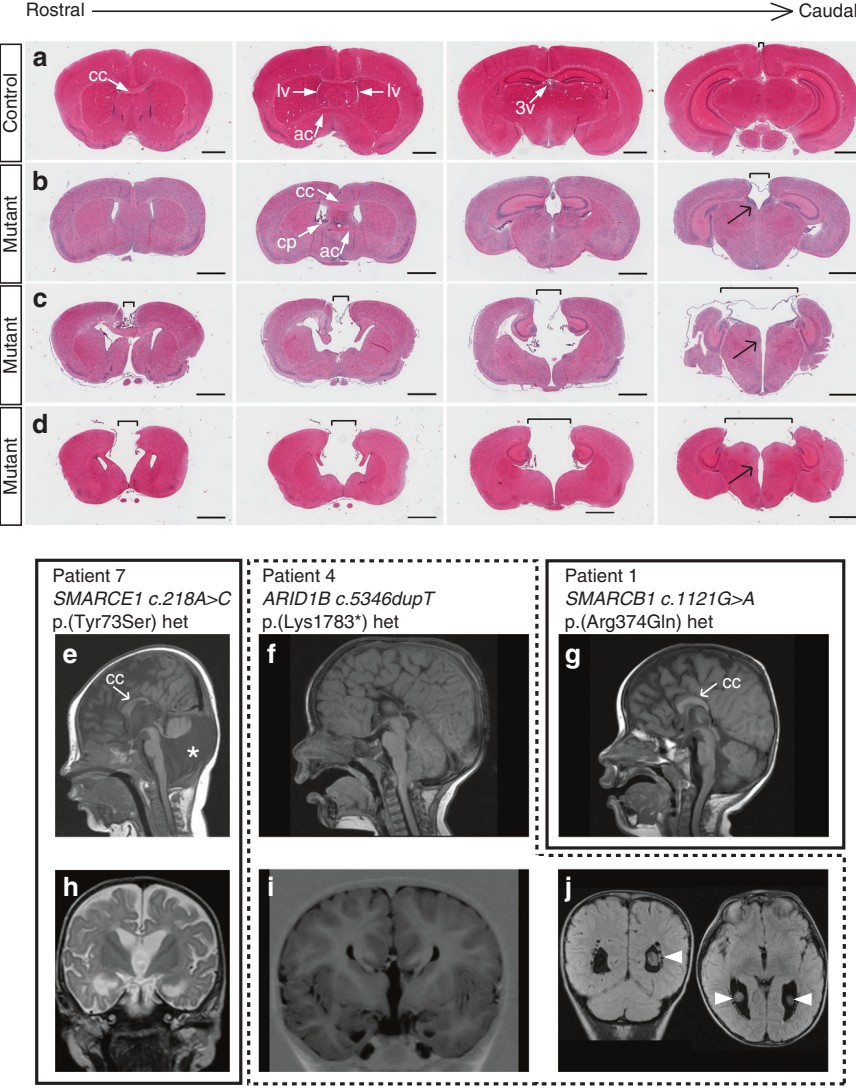

**Fig. 2** Midline brain defects in mutant mice and in Coffin–Siris syndrome (CSS) patients. **a–d** Hematoxylin–eosin-stained coronal sections of 3–4-week-old postnatal animals. One control animal (**a**) and three mutant animals with increasing phenotypic severity (**b–d**) are shown. Note the increased distance between the cerebral hemispheres (brackets) in mutant animals that is most severe in caudal sections at the level of the diencephalon. Black arrows point to a midline cleft between the two halves of the thalamus. ac anterior commissure, cc corpus callosum, cp choroid plexus, lv lateral ventricle, 3v third ventricle. Scale bars: 1 mm. **e–j** Magnetic resonance imaging scans of three CSS individuals with a germline mutation in *SMARCE1* (patient 7, **e**, **h**), *ARID1B* (patient 4, **f**, **i**, **j**), and *SMARCB1* (patient 1, **g**). Note the short and thin CC in **e**, hypoplastic septum pellucidum, small anterior and hippocampal commissures (**h**) and Dandy–Walker variant (asterisk in **e**) with a large posterior fossa, vermis hypoplasia, and cerebellar dysplasia in patient 7. Patient 4 lacks the CC, the anterior and hippocampal commissures, and the septum pellucidum (**f**, **i**). Moreover, a voluminous CP is present (arrowheads in **j**). A short and stubby CC is present in patient 1 (**g**)

**Table 1 Brain MRI findings in CSS individuals**

| No. | Mutation | CC | AC | HC | Septum pellucidum | CP | Cerebellum |
|---|---|---|---|---|---|---|---|
| 1 | *SMARCB1* c.1121G>A p.(Arg374Gln) | **Short stubby** | Not visible | Not visible | Present | Normal | Normal |
| 8 | *SMARCB1* c.1121G>A p.(Arg374Gln) | **Short stubby** | Not visible | Not visible | Present | Normal | Normal |
| 9 | *SMARCB1* c.1091_1093delAGA p.(Lys364del) | **Short** | Normal | Normal | Present | Normal | **Large posterior fossa, hypoplastic vermis, cerebellar hypoplasia** |
| 10 | *SMARCB1* c.1091_1093delAGA p.(Lys364del) | **Short thin** | **Small** | **Small** | Normal | Normal | **Enlarged fourth ventricle, mild vermis hypoplasia** |
| 11 | *SMARCB1* c.1121G>A p.(Arg374Gln) | **Short thin** | Normal | Normal | Normal | Normal | Normal |
| 2 | *ARID1B* c.5961_5964delGAGA p.(Arg1988Ser*fs*\*32) | Normal | Normal | Normal | Present | Normal | Normal |
| 3 | *ARID1B* c.3604_3610dupTCCATGG p.(Ala1204Val*fs*\*8) | **Short stubby** | **Small** | **Small** | Present | Normal | Normal |
| 4 | *ARID1B* c.5346dupT p.(Lys1783*) | **Absent** | **Absent** | **Absent** | **Absent** | **Voluminous** | Normal |
| 5 | *ARID1B* c.2191_2192dupAT p.(Pro732Ser*fs*\*14) | Normal | Normal | Normal | Present | Normal | Normal |
| 6 | *ARID1B* c.5910_5928del p.(Pro1973Arg*fs*\*29) | **Absent** | **Small** | **Small** | **Absent** | Normal | Normal |
| 7 | *SMARCE1* c.218A>C p.(Tyr73Ser) | **Short thin** | **Small** | **Small** | **Hypoplastic** | Normal | **Large posterior fossa, vermis hypoplasia, cerebellar dysplasia** |

Abnormalities are highlighted in bold

*No.* patient number, *CC* corpus callosum, *AC* anterior commissure, *HC* hippocampal commissure, *CP* choroid plexus, *CSS* Coffin–Siris syndrome, *MRI* magnetic resonance imaging

levels were significantly reduced in mutant brain tissue to about 70% of control levels (Fig. 1g).

**CSS individuals show a variety of brain midline defects**. Structural brain abnormalities are seen in the majority of CSS patients[17]; however, a detailed analysis focusing on brain midline defects has not been performed. We systematically re-evaluated brain magnetic resonance imaging (MRI) scans, which were available to us for 11 individuals with CSS. Three of them were previously reported (patients 1[9], 7[9], 10[18]), eight are newly described here (Supplementary Note 1). Of these individuals, nine exhibited brain midline defects of varying severity (Fig. 2e–j, Table 1) and had mutations in *SMARCB1*, *SMARCE1*, or *ARID1B*. MRI scans of two individuals (2 and 5), both with pathogenic variants in *ARID1B*, did not show any structural abnormalities.

**Midline glia and interhemispheric fissure abnormalities**. The development of mammalian brain midline structures involves the invagination of the roof plate. This process not only results in the formation of an interhemispheric space that separates the cerebral hemispheres but is also crucially involved in the development of the CC as the appearance of this commissure is linked to a reduction of this space. The interhemispheric space is initially filled with cells and extracellular matrix and becomes thinner during development, finally resulting in a fissure that separates the closely approximated hemispheres by only a thin layer of meningeal cells at postnatal stages.

To understand the cause of forebrain midline defects in mutant animals, we assessed their brain development. While no anatomical abnormalities were evident at E12.5, we found midline alterations at E15.5 (Supplementary Fig. 3a). Relative to the overall brain dimensions, the interhemispheric distance was increased (Supplementary Fig. 3a) and the thalamic halves were separated by a larger third ventricle space in mutant animals compared to controls (Supplementary Fig. 3a). *Smarcb1* is

ubiquitously expressed, and immunostaining of an E14.5 brain section confirmed the presence of Smarcb1 protein in virtually all nuclei (Supplementary Fig. 3b).

Taken together, our analyses of embryonal and postnatal tissue sections indicate an impaired midline fusion of cerebral hemispheres and of diencephalic regions, particularly in the vicinity of the telencephalic–diencephalic junction, and a failed development of commissures.

To investigate if absent commissures could be due to a failed commissural neuron generation, we analyzed the presence of these neurons by immunostaining. The majority of callosal axons in mice derives from Satb2-expressing neurons in upper cortical layers[19,20]. A large number of these neurons was present in postnatal day 0 (P0) mutant animals and these cells formed a layer that was correctly positioned with respect to Ctip2-expressing deep layer neurons (Supplementary Fig. 4a). However, commissural axons of the CC failed to cross horizontally in mutant mice and instead deviated ventrally at the ipsilateral side (Fig. 3h).

The correct navigation and midline crossing of CC axons are controlled by neuronal and glial guidepost cells. Midline glia populations include the glial wedge (GW), indusium griseum glia (IGG), and midline zipper glia (MZG)[21]. These Gfap-positive, transient cell types disappear postnatally and originate from RGCs[22–24], i.e., neural stem cells that appear from E9 to E10 in the mouse and that give rise to the majority of brain neurons and macroglia[25,26]. GW cells appear at a ventral and lateral location with respect to the developing CC and prevent callosal axons from migrating ventrally[21,22]. IGG demarcate the developing CC dorsally[22]. Defects in their development are associated with CC agenesis[23,27,28]. MZG are detectable below the developing CC[22]. The basement membrane and leptomeningeal cells of the interhemispheric fissure, both of which are recognized by a pan-laminin-antibody, act as barrier for callosal axons to cross the midline[24].

P0 mutant mice exhibited a widened interhemispheric fissure (Fig. 3a–j) that was not crossed by Nrp1-stained callosal axons,

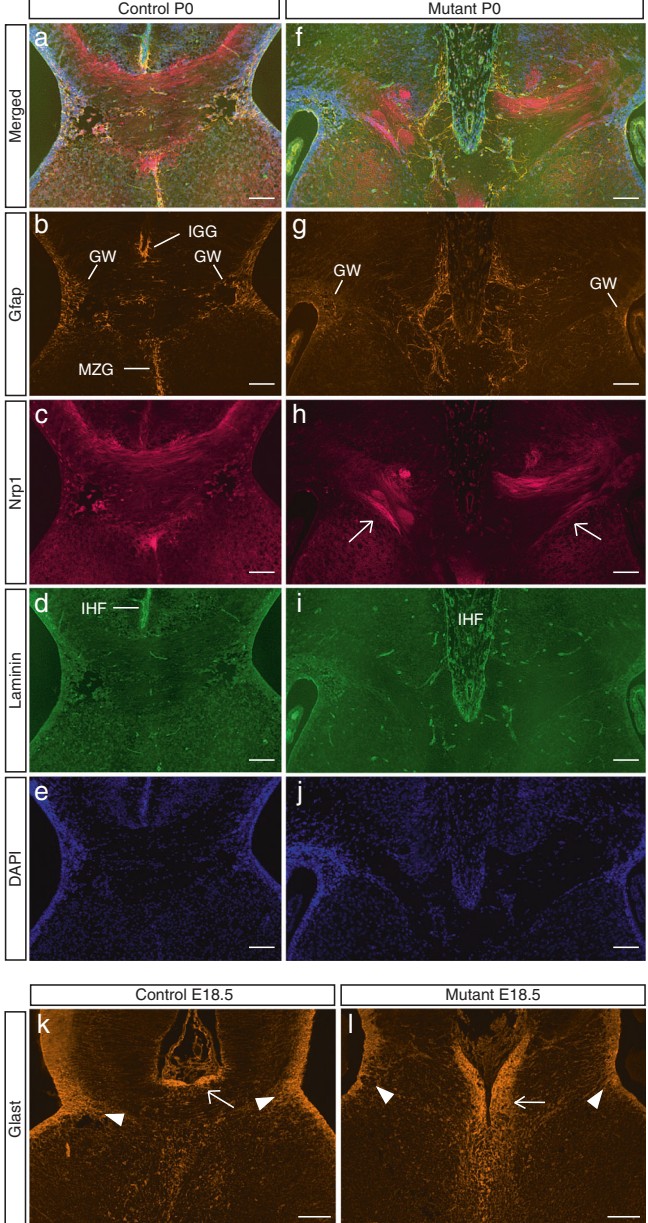

**Fig. 3** Aberrant midline glia in mutant mice. **a–j** Immunostainings of P0 coronal brain sections: Neuropilin 1-positive cortical axons fail to cross horizontally in mutant animals. Instead, some axon bundles deviate ventrally at the ipsilateral side (arrows in **h**). Gfap-positive midline glia are aberrantly positioned and the glial wedge population is diminished in the mutant brain (**g**) GW glial wedge, IGG indusium griseum glia, MZG midline zipper glia. A widened interhemispheric fissure (IHF, laminin-positive connective tissue cells and basement membranes are stained in **d** and **i**) that is not crossed by cortical axons is present in the mutant brain. **k, l** Immunostainings of E18.5 coronal brain sections with an anti-Glast antibody (red immunosignals). The Glast-positive radial glial population at the position of the prospective Gfap-positive GW is reduced and shows shorter radial processes (arrowheads in **k, l**). Glast-positive immunosignals above the developing corpus callosum are horizontally positioned in the control brain (arrow in **k**) but deviate toward the midline in the mutant brain (arrow in **l**). All scale bars: 100 μm

which instead deviated ventrally (Fig. 3h). Gfap-positive glia cells surrounded the fissure in the mutant brain (Fig. 3g) but did not concentrate at the fissure base and along the midline below the CC, and thus did not form distinct IGG and MZG regions as in the control brain (compare Fig. 3b to Fig. 3g). In addition, a reduced Gfap-positive cell population was found at the GW position (Fig. 3g). Interestingly, Glast-positive RGCs, which give rise to Gfap-positive guidepost cells, were already mispositioned at E18.5 (Fig. 3k, l). Furthermore, at the GW position, the area of Glast-positive RGCs and their processes was reduced in the mutant brain (Fig. 3k, l).

Thus the failed midline crossing of commissural axons in mutant mice is associated with an aberrant arrangement of Gfap-positive IGG and MZG guidepost cells and Glast-positive midline radial glial precursor cells at these positions, a paucity of the mature Gfap-positive and the precursor Glast-positive GW populations, and a widening of the interhemispheric fissure.

**The RGC population is diminished in mutant mice**. Nissl stainings of postnatal forebrain sections showed reduced neuronal cell numbers in mutants, which explains the microcephaly (Supplementary Fig. 4b). This finding and the reduced GW population in these animals led us to investigate the number of RGCs and intermediate progenitors, the precursors of neurons and astrocytes in the brain.

Immunostainings of the RGC population in the ventricular zone (Sox2-positive), the intermediate progenitor population in the subventricular zone (Tbr2-positive), and early born (Ctip2-positive) cortical neurons were performed on E14.5 brain sections (Fig. 4a). All three layers were significantly thinned (Fig. 4b), but their cell density was not different (Fig. 4c), indicating that cell numbers of these populations are reduced in the mutant brains. RGCs can be cultured as neurospheres[29], and we found a dramatically decreased neurosphere-forming capacity in cultures from E15.5 mutant brain tissue (Fig. 4d).

Taken together, these data indicate that RGC maintenance is impaired in the mutants, resulting in a reduced number of these cells and their progeny, and thus leading to microcephaly.

**Reduced *Hes5* expression at the developing GW position**. The transcription regulator and Notch effector Hes5 is a key factor for RGC maintenance and promotion of astrocyte development in mice[30]. Interestingly, Notch signaling promotes midline glial cell fates in Drosophila[31] and Smarcb1 binds to *Hes1* and *Hes5* promoters in mouse myoblasts[32]. We investigated the expression of these genes in embryonic brains at the level of developing CC. In situ hybridization experiments did not reveal an obvious difference in *Hes1* expression between control and mutant brains. Interestingly, a distinct *Hes5* pattern was found in the medial walls of the lateral ventricles: in controls, a ventral domain of stronger *Hes5* expression sharply abutted a domain of weaker expression dorsally (Fig. 4e). The border between these low and high *Hes5* expression fields was located at the region where GW RGCs are generated (Fig. 4e). In mutants, the difference between the dorsal and ventral *Hes5* expression was diminished and the expression levels in the ventral region were lower compared with control brains (Fig. 4e).

To conclude, decreased Glast-positive radial glial and Gfap-positive GW cell numbers in the mutant brain are accompanied by reduced *Hes5* expression at the developing GW site. Given the known role of *Hes5* in RGC maintenance and in the promotion of astrocyte formation, our results suggest that diminished *Hes5* activation contributes to these GW defects.

**Increased CP tissue in *Smarcb1* mutant mice**. Increased CP tissue, most strikingly in the lateral ventricles, and also evident in the third ventricle, was another remarkable dorsal forebrain midline alteration in *Smarcb1* mutants (Fig. 5a).

To assess CP development, we performed in situ hybridizations with an antisense probe for *Transthyretin* (*Ttr*), a marker for CP

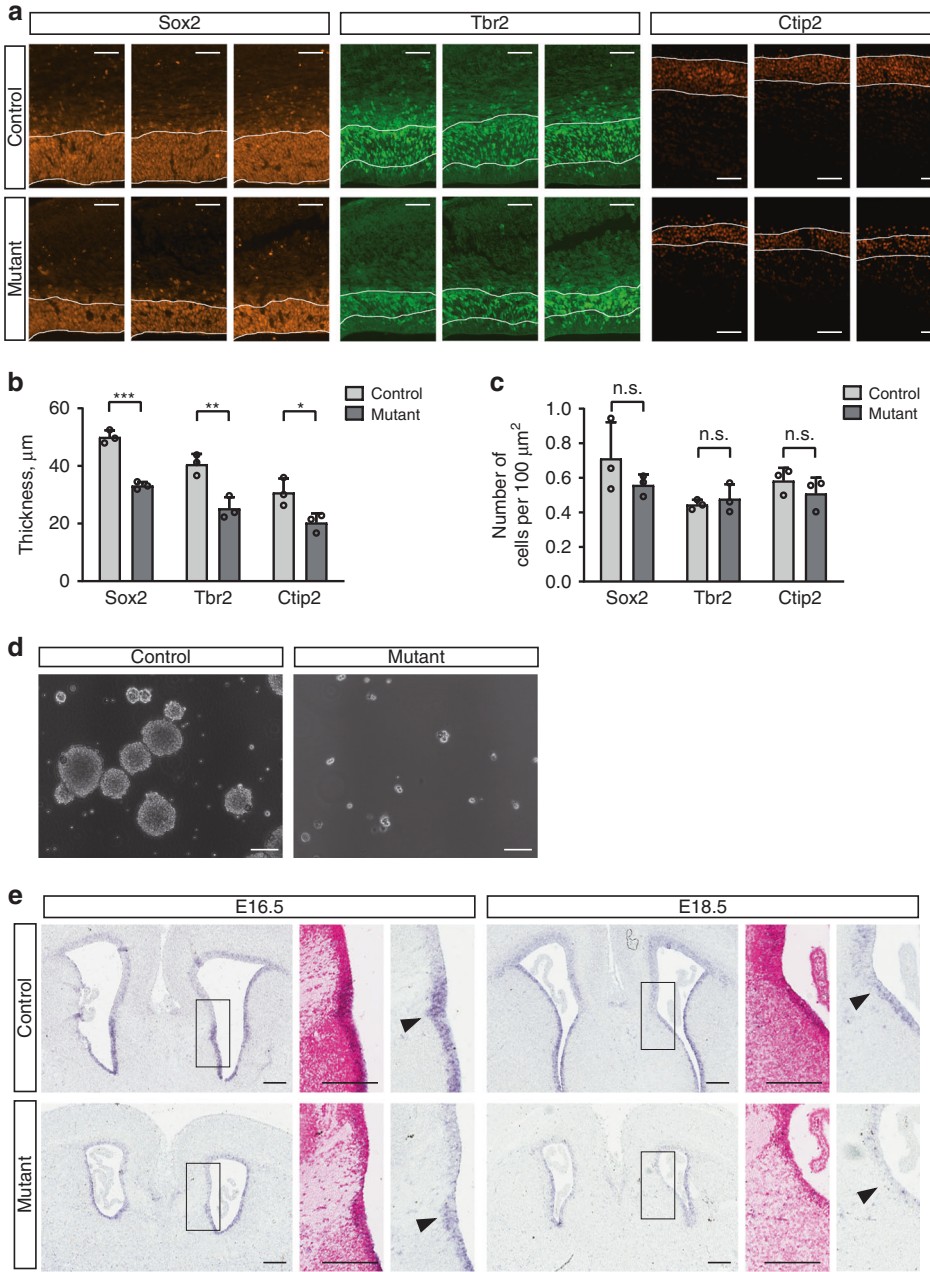

**Fig. 4** Reduced neural stem and progenitor cells in the embryonic mutant brain. **a** Immunostainings of coronal forebrain sections from three mutant and three littermate control embryos at stage E14.5. **b** The thickness of the Sox2-, Tbr2-, and Ctip2-positive cortex layers is significantly reduced in the mutant compared to control animals. All bars display the mean with standard deviations. *$p \leq 0.05$, **$p \leq 0.01$, ***$p \leq 0.001$ (unpaired two-tailed Student's $t$ test). **c** Cell densities within the Sox2-, Tbr2-, and Ctip2-positive layers of the mutant compared with the control cortex. $p$ values (unpaired two-tailed Student's $t$ test): n.s. not significant. **d** Cells isolated from the mutant forebrain of E15.5 embryos fail to form large neurospheres in comparison with cultures established from E15.5 control brain tissue. Cultures of passage one (3 days after passaging) are shown. **e** *Hes5* expression in coronal brains sections from E16.5 and E18.5 mutant and control animals detected by in situ hybridization (blue signals). Enlarged regions (framed areas) include the site of glial wedge development (arrowheads). Nuclear fast red stainings were performed to visualize the region of corpus callosum development, which appears white due to the paucity of nuclei. Scale bars: 100 μm (**a**, **d**), 200 μm (**e**). Source data are provided as a Source Data file

epithelium (Fig. 5a, b). These experiments revealed no alteration in early telencephalic CP field specification in the mutants but an increased lateral and third ventricle CP tissue mass from E16.5.

Next, we investigated CP tissue growth. In mutants and controls, cells producing Ki67, a protein present during all cell cycle phases but absent in $G_0$, were found in the proximal part of the lateral ventricle CP epithelium at E14.5 and P0 (Fig. 5c). The presence of dividing cells at this position, i.e., near the root of the CP, is in accordance with ref. [33]. In 4-week-old mutants and controls, no Ki67-positive epithelial CP cells were present any longer. The number of dividing epithelial cells was not significantly higher in the proximal part of the lateral ventricle CP in E14.5 and P0 mutant animals (Fig. 5d). However, the number of CP branches was increased in the mutant lateral ventricle at P0 (Fig. 5c, e), which can also be seen on *Ttr* in situ hybridization sections (lowest panels in Fig. 5a).

In summary, mutant mice exhibited CP enlargement in the lateral and third ventricles, which seems to be due to increased

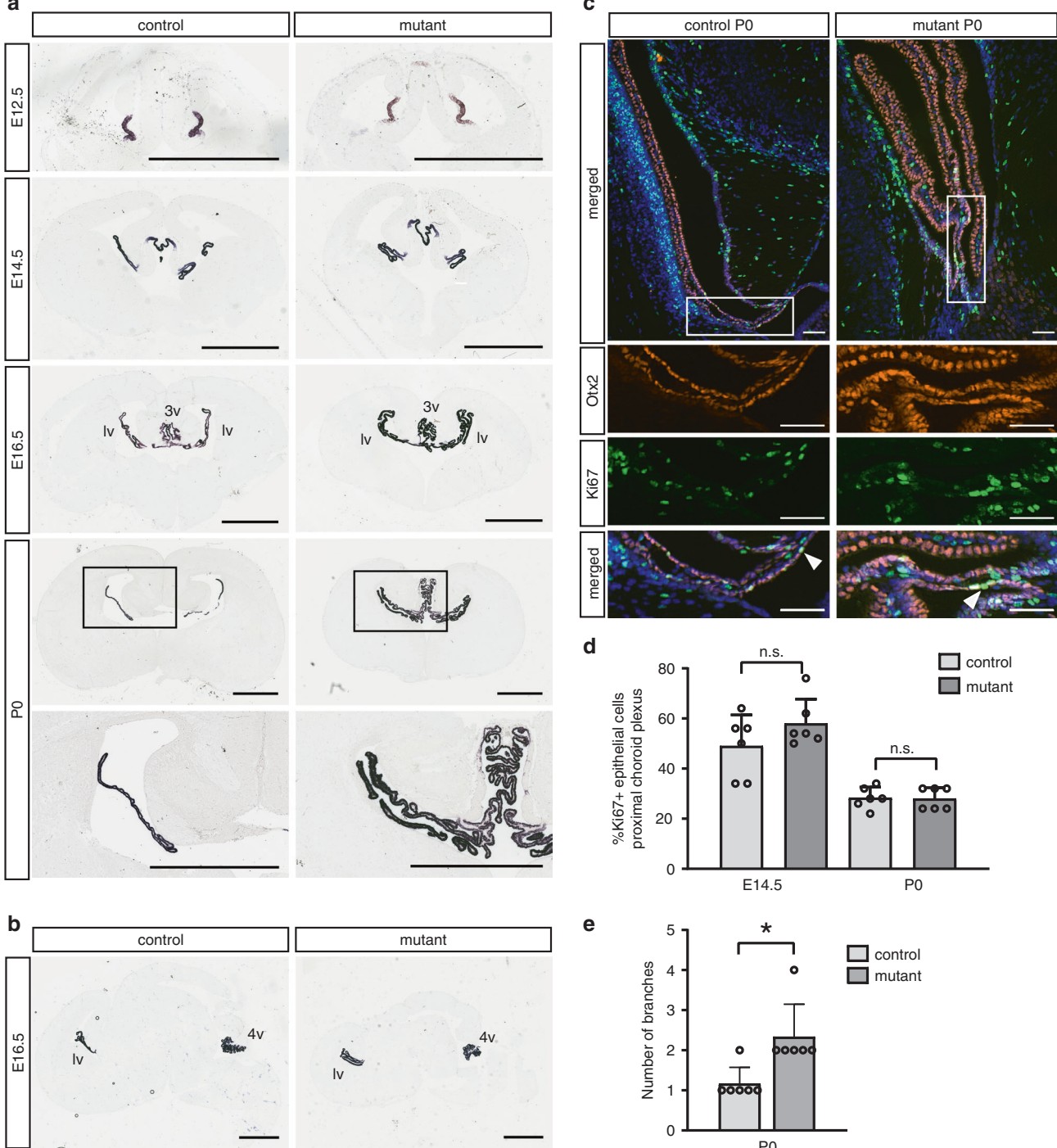

**Fig. 5** Mutant mice develop a larger mass of *Ttr*-expressing choroid plexus (CP). **a**, **b** *Ttr* expression detected by in situ hybridizations. **a** Coronal brain sections at the level of the lateral to third ventricle junction from different developmental stages. Note the increased number of CP tissue folds in the lateral and third ventricles of the P0 mutant brains. Enlarged regions of P0 sections are shown at the bottom. **b** Sagittal sections from E16.5 animals. Scale bars (**a**, **b**): 1 mm. **c** Immunostainings of P0 coronal brain sections containing the site of CP attachment to the ventricle wall (proximal CP). Otx2-positive CP epithelial cells (red), Ki67-positive cells (green), and DAPI-positive nuclei (blue) are shown. Lower panels display higher magnifications of the proximal CP region framed in the upper panel. Arrowheads point to Ki67-positive CP epithelial cells. Scale bars: 100 μm. **d** Percentage of Ki67-positive cells present in the proximal Otx2-positive CP tissue determined in E14.5 and P0 animals. The most proximal 50 cells starting from the attachment point of the CP were analyzed. The right and the left lateral ventricle of three control and three mutant animals of the same litter were investigated (six choroid plexi per group). Bars display the mean with standard deviations. *p* values (unpaired two-tailed Student's *t* test): n.s. not significant. **e** Number of branches in CP tissue of P0 animals. The right and the left lateral ventricle of three control and three mutant animals of the same litter were investigated (six choroid plexi per group). Bars display the mean with standard deviations. *p* values (unpaired two-tailed Student's *t* test): *\*p* ≤ 0.05. Source data are provided as a Source Data file

branching of this tissue. Based on histological criteria (intact papillary architecture, no nuclear pleomorphisms, no necrosis, no increased proliferation at postnatal stages), we classify the enlarged CP tissue as benign lesion. Considering human CP pathologies, these alterations resemble diffuse villous hyperplasia or bilateral CP papilloma (the latter is World Health Organization (WHO) grade I). It is currently unresolved if these are two distinct entities and no clear diagnostic criteria exist to distinguish them[34].

***Smarcb1* mRNA levels in embryonic *Smarcb1*[+/−] brain tissue.** In contrast to *Smarcb1* mutant mice described here, no abnormalities in terms of body size, head size, or brain anatomy have been reported for heterozygous conventional *Smarcb1* knockout mice (*Smarcb1*[+/−]) generated by three different groups[13,35,36]. The only phenotype ascribed to these animals are soft tissue tumors, mainly sarcomas, that are thought to develop upon additional *loss-of-function* of the second *Smarcb1* allele.

To better understand why a complete inactivation of one *Smarcb1* allele, which takes place in all cells from the zygote stage on (*Smarcb1*[+/−]), does not lead to the phenotype seen in *Smarcb1*[+/inv] *NesCre*[+/−] mice, we investigated heterozygous *Smarcb1*[+/−] animals generated by ref. [13] (Supplementary Fig. 5). Brains from nine investigated postnatal *Smarcb1*[+/−] mice were macroscopically indistinguishable from their *Smarcb1*[+/+] littermates and sections of six analyzed animals did not reveal any brain midline defects (Supplementary Fig. 5c). In contrast to reduced transcript levels in E14.5 forebrain tissue from *Smarcb1* mutant animals, quantitative reverse transcriptase-polymerase chain reaction (qRT-PCR) analyses revealed no significant difference in *Smarcb1* transcript abundance between *Smarcb1*[+/−] and *Smarcb1*[+/+] littermates, also in case of primers that should only amplify a sequence from the wild-type allele in *Smarcb1*[+/−] tissue (Supplementary Fig. 5d). Thus the lack of a neurodevelopmental phenotype in *Smarcb1*[+/−] animals can be explained by unaltered *Smarcb1* transcript levels, which are due to a compensatory upregulation of *Smarcb1* gene expression from the second wild-type allele. This mechanism has been analyzed in embryonic fibroblasts from these mice[37].

## Discussion

In comparison to other transgenic mice with midline glia aberrations, several aspects of the *Smarcb1* mutant animals are unique, indicating that the molecular consequences of reduced *Smarcb1* levels are not identical to the genetic alterations in other models. To our knowledge, a large forebrain midline cleft, as seen in the severely affected *Smarcb1* mutants, has not been found in any other published model with midline glia aberrations and/or forebrain commissural defects, nor do other mouse models display additional midline defects seen in *Smarcb1* mutant mice, namely, increased CP tissue and cerebellar midline aberrations. For example, transgenic mice with a conditional inactivation of *Lhx2* in cortical RGCs (*Cre* expressed under *Emx1* or *Nestin* promoter elements) lack the hippocampal commissure and CC and show a prominent cleft between the cerebral hemispheres[38]. In contrast to *Smarcb1* mutants, however, this cleft does not further extend ventrally and all animals possess an anterior commissure. Interestingly, the CC defect in *Lhx2* mutants is thought to be caused by a reduced to absent GW due to abnormal radial GW precursors, a feature similar to the *Smarcb1* mutant mice. Animals lacking transcription factors Nfia or Nfib are characterized by a failure of precursor cells to mature to Gfap-positive midline astroglia[24,39–42]. In *Smarcb1* mutant mice, however, Gfap-positive midline glia are present, but their localization is disturbed and the GW population is diminished. The defect in homozygous *Nfia* or *Nfib* knockout mice is accompanied by an abnormal retention of the interhemispheric fissure. This fissure is, however, much thinner and does not result in a cleft as in *Smarcb1* mutants[24]. Smarcb1 is a component of chromatin remodeling complexes that modulate chromatin state and DNA accessibility to transcriptional regulators, thereby influencing gene expression patterns. In the present study, we identified reduced *Hes5* transcript levels at the site of GW development, and it will be important to find out if the various abnormalities in different brain regions of *Smarcb1* mutant mice are all caused by the same or different downstream consequences of reduced *Smarcb1* levels in RGCs.

Although single clinical reports on structural brain abnormalities in CSS exist, a thorough and comparative analysis of brain midline defects has not been published, and the developmental basis of these defects is unknown.

We have generated the first *Smarcb1* mutant mouse model that shows features of *SMARCB1*-CSS and *SMARCB1*-related ID-CPH and exhibits a benign lesion of increased tissue mass (CP hyperplasia/papilloma) (Table 2). These mutant mice have striking brain midline defects affecting (i) the telencephalon (lack of the hippocampal commissure, partial or complete lack of the CC and the anterior commissure, disturbed fusion of cerebral hemispheres, and increased lateral ventricle CP tissue), (ii) the diencephalon (disturbed fusion of thalamic halves, increased third ventricle CP), and (iii) the cerebellum (fusion of the cerebellar hemispheres and hypoplastic vermis-like structure). Interestingly, a brain midline defect, namely, decreased CC volume, is also present in animals with a heterozygous *loss-of-function* mutation in another CSS-causing gene, *Arid1b*[43].

**Table 2 Comparison of mutant mice with SMARCB1-related CSS and ID-CPH**

| | CSS patients with proven *SMARCB1* mutations | ID-CPH patients with a distinct *SMARCB1* mutation | Mutant mice (*Smarcb1*[+/inv] *NesCre*[+/−]) |
|---|---|---|---|
| Microcephaly (in patients: reduced occipitofrontal circumference) | (15/15) 100% | (0/4) 0% | 100% |
| Agenesis or hypoplasia of the corpus callosum | (14/15) 93% | (4/4) 100% | 94% |
| Cerebellar hypoplasia | (2/15) 13% | (2/4) 50% | 100% |
| Cerebellar midline abnormality (in patients: Dandy–Walker malformation or variant, vermis hypoplasia) | (4/15) 27% | Reduced vermis size (2/4) 50% | 95% |
| Choroid plexus hyperplasia | Not reported/found | (4/4) 100% | 100% |
| Growth impairment | (16/16) 100% | Intrauterine growth retardation (1/4) 25% | 100% |

Data on CSS individuals are taken from Kosho et al. (2014)[17], Gossai et al. (2015)[51], and individuals 1 and 8–11 described in this article; data on ID-CPH individuals are taken from Diets et al. (2018)[11].
*CSS* Coffin–Siris syndrome, *ID-CPH* intellectual disability-choroid plexus hyperplasia

We identified similar midline defects, reduced to absent telencephalic commissures, hypoplastic, or absent septum pellucidum, and increased lateral ventricle CP tissue, in 9 of the 11 CSS patients (5 with *SMARCB1* mutations, 1 with a *SMARCE1* mutation, and 3 with *ARID1B* mutations), although not all of these defects were present in every patient. A cerebellar midline abnormality was found in three cases: mild vermis hypoplasia in an individual with a *SMARCB1* mutation, and a Dandy–Walker variant in two individuals, one with a *SMARCE1* mutation and one with a *SMARCB1* mutation. Developmental cerebellar defects defined as the Dandy–Walker syndrome are classified into three categories (variant, malformation, Mega cisterna magna), all of which can occur in CSS. The variant is characterized by a hypoplastic cerebellar vermis and a mild enlargement of the fourth ventricle and posterior fossa. The cerebellar midline defect in mutant mice includes hypoplasia of a vermis-like structure and fusion of the cerebellar hemispheres. The former is a feature of the Dandy–Walker spectrum, the latter is reminiscent of rhombencephalosynapsis, which is also associated with partial or complete vermis agenesis; rhombencephalosynapsis itself has so far not been reported in CSS.

The brain midline defects in these nine patients have so far not been recognized as a characteristic feature of CSS. This might be due to different reasons: (i) brain MRIs are not always performed, (ii) only very obvious aberrations such as CC agenesis are typically diagnosed, and (iii) the Dandy–Walker syndrome phenotypes are less frequent in CSS. Moreover, CC and cerebellar alterations were not seen as potentially associated with each other. Kosho et al. found that, among CSS patients with analyzed structural brain abnormalities, the majority had CC agenesis (eight of nine with *SMARCB1*, six of seven with *SMARCA4*, one of two with *SMARCE1*, and seven of eight with *ARID1A* mutations)[17]. Moreover, a Dandy–Walker malformation was present in one of nine (*SMARCB1*), two of seven (*SMARCA4*), one of two (*SMARCE1*) and one of eight (*ARID1A*) cases[17]. A survey on CSS and ID individuals with pathogenic *ARID1B* variants revealed a high frequency of complete CC agenesis (29% in ARID1B-CSS and 28.2% in ARID1B-ID) and hypoplasia/partial CC agenesis (17.7% in ARID1B-CSS and 7.7% in ARID1B-ID)[44]. A thin CC is also present in all four reported patients with *SMARCB1*-related ID-CPH[11]. In two of these, a reduced vermis size was found.

It is currently not understood how alterations in genes encoding BAF complex components can lead to clinical outcomes that were so far recognized as distinct disease entities: tumor development or ID/neurodevelopmental disorders[3].

*SMARCB1* mutations in CSS are germline de novo heterozygous non-truncating missense mutations or small in-frame deletions within exons 8 and 9[6,7,9]. Patients with *SMARCB1*-related ID-CPH carry a de novo germline missense mutation in exon 2 (c.110G>A;p.Arg37His)[11]. The effect of these mutations on the function of the SMARCB1 protein remains unknown. *SMARCB1* heterozygous germline mutations underlying tumor predisposition syndromes, the rhabdoid tumor predisposition syndrome 1 (RTPS1) and schwannomatosis, are in the vast majority of a different type. RTPS1 patients carry germline heterozygous truncating *loss-of-function* mutations or large deletions, and somatic loss of heterozygosity leads to malignant intracranial and extracranial rhabdoid tumors as well as other malignancies such as CP carcinomas, medulloblastomas, and central primitive neuroectodermal tumors[45]. No SMARCB1 protein is typically detectable in these tumor tissues. In contrast, the other SMARCB1 tumor predisposition syndrome, schwannomatosis, is typically caused by heterozygous non-truncating germline *SMARCB1* mutations that mostly occur in exons 1, 2 or in the 3′-untranslated region (3′UTR)[46,47]. Affected patients develop benign tumors, nerve sheath tumors (schwannomas), and

meningiomas. Frequent loss of the second allele in combination with biallelic loss of *NF2* has been reported in such tumor tissues, indicating that SMARCB1 acts as a tumor suppressor in schwannomas and that the familial *SMARCB1* mutations lead to a *loss-of-function*. Analyses of such mutations suggest that they do not result in a complete loss of the protein as in rhabdoid tumors but in reduced *SMARCB1* gene expression[46,47]. The occurrence of either malignant rhabdoid tumors or schwannomatosis in families with germline *SMARCB1* mutations[48,49] and the occurrence of schwannomatosis in a patient that survived a malignant rhabdoid tumor[50] support the idea that mutations in both RTPS1 and schwannomatosis are *loss-of-function* mutations of different severity but with an overlapping phenotypic spectrum.

To date, a single CSS individual with a germline missense *SMARCB1* mutation has been reported to suffer from a *SMARCB1*-related tumor entity, namely schwannomatosis, a benign tumor disease[51]. In this patient with a *SMARCB1* mutation in exon 9, inactivation of the second allele had occurred in the tumor tissue. Strikingly, the co-occurrence of ID and a benign overgrowth lesion (CP papilloma/hyperplasia) is described in four unrelated individuals with a recurrent germline missense mutation in *SMARCB1* exon 2 (SMARCB1-related ID-CPH)[11]. These patients show some overlap with CSS individuals (brain midline abnormalities, severe ID, congenital heart defects, kidney abnormalities, and feeding difficulties) but lack other features that are typical for *SMARCB1*-CSS (impaired growth, microcephaly, fifth finger anomalies, scoliosis, and epilepsy) (Table 2).

*SMARCB1* mutations occur in a substantial proportion of CP tumors[52–56]. The majority are carcinomas (WHO grade III) and show *SMARCB1* mutations that are predicted to result in a truncated protein. It remains unclear whether some or potentially all of the *SMARCB1*-mutated malignant CP tumors (carcinomas) represent a separate entity or are misdiagnosed atypical teratoid/ rhabdoid tumors[52,57]. *SMARCB1* mutations have been found in two CP papilloma[58] and two atypical CP papilloma (WHO grade II) cases[53,58]. The papillomas and one atypical papilloma tumor are possibly due to homozygous deletions encompassing exon 9[58]; an acquired homozygous deletion including exon 4 was found in the other atypical papilloma[53].

The *Smarcb1* mutant mouse model combines features of both neurodevelopmental defects and a benign lesion of increased tissue mass (CP hyperplasia/papilloma). Several abnormalities found in four individuals with *SMARCB1*-related ID-CPH[11] are recapitulated by this model (Table 2): CP enlargement in both lateral ventricles that occurs early during development (diagnosed prenatally or early postnatally in patients), CC abnormalities (thinned in all four cases), and reduced cerebellar vermis (mild atrophy in one case and small size in another case). However, microcephaly and growth impairment, two features typically associated with *SMARCB1*-CSS and present in all *Smarcb1* mutant mice, are not characteristic for these four patients.

The genetic alteration in *Smarcb1* mutant animals (monoallelic reversible disruption of exon 1 in *Nestin*-expressing cells), and the fact that *Smarcb1* mRNA levels were reduced by about 30% in embryonic brain tissue, indicate that this phenotype is due to a partial *loss-of-function*. Notably, *Smarcb1* mRNA levels were unaltered in embryonic brain tissue from *Smarcb1*[+/−] mice, which permanently lack one allele and do not show obvious brain abnormalities. A potential mechanism underlying the compensatory increase of transcripts from the wild-type allele in these mice is a negative autoregulation of the *SMARCB1* gene. Moreover, the permanent monoallelic inactivation of *Smarcb1* in *Nestin*-expressing cells in a mouse model with *Smarcb1* exon 1 flanked by loxP sites of identical orientation (*Smarcb1*[+/flox] *NesCre*[+/−]) does not lead to an apparently aberrant brain phenotype either[59]. These results suggest that the complete monoallelic *Smarcb1* inactivation

in $Smarcb1^{+/-}$ and $Smarcb1^{+/flox}$ $NesCre^{+/-}$ mice can be fully compensated by an increased expression from the wild-type allele, whereas the cellular compensatory mechanisms do not suffice to counterbalance the effect of the continuous exon 1 inversion in $Smarcb1^{+/inv}$ $NesCre^{+/-}$ mice.

In light of our data and the above-mentioned findings, it is tempting to interpret the consequences of pathogenic germline *SMARCB1* alterations according to three categories.

i. Monoallelic alterations resulting in a partial SMARCB1 *loss-of-function* due to reduced *Smarcb1* mRNA levels (as in our mouse model) or missense mutations leading to the generation of an altered SMARCB1 protein product (as in CSS and ID-CPH patients) cannot be fully compensated by the second intact allele and lead to ID/neurodevelopmental defects and to benign brain lesions (CP hyperplasia/papilloma).

ii. Complete *loss-of-function* alterations in one allele (truncating mutations and large deletions) that lead to a complete lack of a protein product can be fully compensated through an increased expression from the second intact allele and do not lead to neurodevelopmental defects. However, they predispose to malignant tumors upon *loss-of-function* of the second allele.

iii. Germline monoallelic mutations predisposing to schwannomatosis (benign tumors) are typically non-truncating and include missense and nonsense mutations, splice site alterations, and mutations in the 3′UTR. Such mutations have never been detected in CSS and some of them are predicted to lead to reduced *SMARCB1* expression. The effect of these mutations could either be fully compensated or not suffice to cause a CSS/neurodevelopmental phenotype. Conversely, a CSS-causing mutation in *SMARCB1* exon 9 can predispose to schwannomatosis. Moreover, rhabdoid tumor-predisposing mutations can also lead to schwannomatosis. In addition to the type of mutation, the developmental stage and thus the cell type in which two mutated alleles are present is another determinant of schwannoma versus rhabdoid tumor development[60].

So far, it was not understood how mutations in *SMARCB1* lead to any of the features of CSS and ID-CPH. Our data highlight a novel role of *Smarcb1* in brain midline development, aspects of which can be explained by a reduced number and mispositioning of midline RGCs. Moreover, our results are of high clinical relevance. We suggest that bilateral CP hyperplasia/papilloma should be regarded as indicative for potential mutations in a CSS-causing gene. Finally, we show that the spectrum of brain midline defects in CSS is broader than so far recognized, thereby providing novel, highly relevant classification criteria for CSS and related ID/neurodevelopmental disorders.

## Methods

**Patients**. Eleven patients with a molecularly confirmed diagnosis of CSS were re-evaluated for brain malformations. Cranial MRI scans were systematically reviewed for structural brain abnormalities by a pediatric radiologist (J.S.). Patients 1, 7, and 10 were described previously[9,18]. Data on the other patients have not been published yet. Some cases were part of larger studies approved by local institutional review boards (ethical votum 12-5089-BO for CRANIRARE and 5590 for Chromatin-Net; ethics committee of Ärztekammer Hamburg, PV3802; ethics committee at the Warsaw Medical University). Written informed consent to publish patient data was obtained from all families of the index individuals. The study was performed according to the Declaration of Helsinki protocols.

**Animals**. $Smarcb1^{+/inv}$ mice, originally termed $Snf5^{+/inv}$, were kindly provided by Charles W. Roberts (Children's Hospital Boston, Harvard Medical School, Boston, USA). We confirmed that the loxP sites surrounding *Smarcb1* exon 1 are of opposite orientation by sequencing genomic DNA from these animals. $NesCre^{+/-}$ mice were kindly provided by Günther Schütz (German Cancer Research Center,

Heidelberg, Germany). $Smarcb1^{+/-}$ mice were a kind gift from Stephen N. Jones (University of Massachusetts Medical School, Worcester, USA). Alleles were maintained on a C57BL/6N background. All animal procedures were approved by the local authorities (Regierungspräsidium Darmstadt, Hesse, Germany). To confirm developmental stages of mouse embryos, anatomical criteria were applied, including the morphology of the limb buds/limbs and the cephalic region.

**Neurosphere culture**. E15.5 forebrain tissue was minced and incubated with Accutase (PAA) for 15 min at 37 °C. The cell suspension was triturated with a pipet tip, passed through a 100 μm cell strainer and centrifuged at $200 \times g$ for 3 min. The pellet was resuspended in serum-free neurosphere proliferation medium (Dulbecco's modified Eagle's medium/F12 medium (Gibco) supplemented with B27 supplement (Gibco), 30 mM HEPES (Gibco), 20 ng/mL epidermal growth factor (R&D), and 20 ng/mL fibroblast growth factor (R&D)). Six days after the primary tissue isolation, cells were passaged using Accutase and cultured in neurosphere proliferation medium at a density of $10^5$ cells/mL.

**PCR using genomic DNA as template**. Genomic DNA was isolated using the E.Z. N.A. Tissue DNA Kit (OMEGA bio-tek, USA). Standard genotyping PCR conditions using ear tissue (final reaction volume of 25 μL, final primer concentrations 0.4 μM) were as follows. Primer sequences are listed below in Table 3.

$Smarcb1^{inv}$: Primers Snf5CE2251 and Snf5CER2454; 95 °C 5 min, 25 cycles (95 °C 1 min, 66 °C 1 min [−0.4 °C per cycle], 72 °C 1 min), 20 cycles (95 °C 1 min, 56 °C 1 min, 72 °C 1 min), 72 °C 10 min.

*NesCre*: Primers Cre fw4, Cre rev4, Gabra1 fw, Gabra1 rev; 95 °C 5 min, 35 cycles (95 °C 30 s, 62 °C 30 s, 72 °C 1 min), 72 °C 5 min.

$Smarcb1^-$: Primers B-Geo fw2, B-Geo rev2, Gabra1 fw, Gabra1 rev; 95 °C 5 min, 35 cycles (95 °C 30 s, 59 °C 30 s, 72 °C 30 s), 72 °C 5 min. Gabra1 primers served as internal positive control.

To determine *Smarcb1* knockout alleles in brain tissue, the following reactions were performed. $Smarcb1^-$: Primers B-Geo fw, B-Geo rev, Smarcb1 exon 7 fw, Smarcb1 exon 7 rev; 95 °C 5 min, 35 cycles (95 °C 30 s, 62 °C 30 s, 72 °C 30 s), 72 °C 5 min. $Smarcb1^{inv}$: Primers Smarcb1 A, Smarcb1 B, Smarcb1 C; 95 °C 5 min, 35 cycles (95 °C 30 s, 58 °C 30 s, 72 °C 30 s), 72 °C 10 min.

**Staining of tissue sections**. Tissue was fixed in 4% formaldehyde/phosphate-buffered saline (PBS) overnight, embedded in paraffin, and 5 μm sections were cut using a Jung RM2055 device (Leica Biosystems) and placed onto Thermo Scientific™ Polysine Adhesion slides. Prior to stainings, sections were deparaffinized and rehydrated. For Nissl stainings, sections were incubated in 0.5% cresyl violet in 133 mM sodium acetate, pH 4.0 for 10 min at room temperature (RT) and washed twice with the sodium acetate solution for 5 min. Sections were washed with

## Table 3 List of primers

| Name | 5′–3′ sequence |
| --- | --- |
| B-Geo fw | CGGTATCGATAAGCTTGATGATCTG |
| B-Geo rev | CGCATCGTAACCGTGCATCTGC |
| B-Geo fw2 | TATCCGAACCATCCGCTGTG |
| B-Geo rev2 | GTGGCCTGATTCATTCCCCA |
| Cre fw4 | GGTTCGCAAGAACCTGATGG |
| Cre rev4 | GCCTTCTCTACACCTGCGG |
| Gabra1 fw | AACACACACTGGAGGACTGGCTAGG |
| Gabra1 rev | CAATGGTAGGCTCACTCTGGGAGATGATA |
| Smarcb1 A | GAAAATCTAGAAAGCACAAATGAGAG |
| Smarcb1 B | CAGGAAAATGGATGCAACTAAGAT |
| Smarcb1 C | GCCACATAAACTGGGTGTAG |
| Smarcb1 exon 1 fw | ATGGCGTTGAGCAAGACC |
| Smarcb1 exon 2 rev | CATACGCAGGTAGTTTCCCAC |
| Smarcb1 exon 3 fw | TGGATGGCAATGACGAGAAG |
| Smarcb1 exon 4 rev | ATGCGGTTCCTGTTGATGG |
| Smarcb1 exon 5 fw | CGAGACGCTTTTACCTGGAAC |
| Smarcb1 exon 6 rev | GGGGTAGGACTCAATCTGCT |
| Smarcb1 exon 7 fw | CTGAACATCCACGTGGGGAACATC |
| Smarcb1 exon 7 rev | CTGAAGGCATAGGTCTTCTGGTGC |
| Smarcb1 isoform 1 fw | ACAAAACCTAACACTAAGGATCATG |
| Smarcb1 isoform 1 rev | CTTCTCGTCATTGCCATCCA |
| Smarcb1 isoform 2 fw | CTACATGATCGGCTCCGAG |
| Smarcb1 isoform 2 rev | TATCCATGATCATGTGACGATGC |
| Snf5CE2251 | CACCATGCCCCCACCTCCCCTACA |
| Snf5CER2454 | CAGGAAAATGGATGCAACTAAGAT |
| Tbp fw1 | GGGAGAATCATGGACCAGAA |
| Tbp rev1 | TTGCTGCTGCTGTCTTTGTT |

distilled water, dehydrated gradually in ethanol (70%, 80%, 90%, and 100%), and mounted in Entellan. For hematoxylin and eosin staining, slides were incubated in hematoxylin for 3–5 min, in tap water for 2 min, stained with eosin Y for 4 min, rinsed with water, dehydrated gradually in ethanol (70%, 80%, 90%, and 100%, 30 s each step), and mounted in Entellan. Images were acquired using an Aperio CS2 slide scanner (Leica Biosystems). For immunostainings, an epitope retrieval was performed by boiling sections in citrate buffer (10 mM Sodium citrate, 0.05% Tween-20, pH6.0) for 40 min. After a permeabilization step (0.5% Triton X-100/PBS for 10 min), primary antibodies were applied in 0.01% Triton X-100/PBS overnight at 4 °C. 0.01% Triton X-100/PBS only served as negative control. Sections were washed with PBS three times (10 min each) and incubated with secondary antibodies diluted in 0.01% Triton X-100/PBS at RT for 1 h followed by washing in PBS (10 min, two times). Specimens were then incubated in DAPI solution (1 μg/mL, Roche) for 10 min, washed in PBS (5 min), rinsed briefly in water, dehydrated in ethanol, and air dried. For mounting, Fluorescence Mounting Medium (DAKO/Agilent) and coverslips (Thermo Scientific, Germany) were used. Immunofluorescent images were taken with a Zeiss Observer D1 or Zeiss Axiovert 200M microscope. The following primary antibodies were applied: Ctip2 (Abcam, ab18465, 1:500), Gfap (Dako, Z0334, 1:800), GLAST (Abcam, ab416-200, 1:100), Ki67 (Abcam, ab16667, 1:100), Laminin (Sigma, L9393, 1:400), Nrp1 (R&D, AF566, 1:100), Otx2 (R&D, AF1979, 1:1000), Satb2 (Abcam, ab69995, 1:200, kind gift from B. Schuetz, University of Marburg, Germany), SMARCB1 (CST, 91735, 1:500), Sox2 (Santa Cruz Biotechnology, sc-17320, 1:150), Tbr2 (Abcam, ab23345, 1:250). Secondary antibodies coupled to Cy3, AF488, and AF647 were from Dianova and diluted 1:300.

**Quantitative RT-PCR**. For cDNA synthesis, 1 μg of total RNA (isolated using the AllPrep DNA/RNA Mini Kit, Qiagen) was incubated with 1 μL random primers (500 ng/μL, Thermo Scientific) in a 12.5 μL reaction volume at 65 °C for 5 min followed by a short incubation on ice and reverse transcription at 42 °C for 60 min upon addition of 200 U M-MuLV reverse transcriptase (Thermo Scientific), 10 U RNAsin (Thermo Scientific), 2 μL 10 mM dNTPs (Thermo Scientific) and 4 μL 5x M-MLV reaction buffer Thermo Scientific) to a final reaction volume of 20 μL. StepOnePlus System (Applied Biosystems) and SYBR Green as fluorescent reporter were used for qRT-PCR measurements. Gene-specific PCR products were measured continuously, and the difference in the threshold number of cycles between the gene of interest and TATA-binding protein (Tbp) was then normalized relative to the standard chosen for each experiment and converted into fold difference. Four μL of 1:100 diluted cDNA were used as template in a 25 μL qPCR reaction with 12.5 μL of Power Up SYBR Green Master Mix (Applied Biosystems) and 0.2 μM of the respective primers. qPCR reaction conditions were as follows: 95 °C 10 min, 40 cycles (95 °C 15 s, 55 °C 1 min). Primer combinations for Smarcb1 exon 1–2 amplification were: Smarcb1 exon 1 fw and Smarcb1 exon 2 rev; for Smarcb1 isoform 1: Smarcb1 isoform 1 fw and Smarcb1 isoform 1 rev; for Smarcb1 isoform 2 amplification: Smarcb1 isoform 2 fw and Smarcb1 isoform 2 rev; for Smarcb1 exon 3–4 amplification: primers Smarcb1 exon 3 fw and Smarcb1 exon 4 rev; for Smarcb1 exon 5–6 amplification, primers Smarcb1 exon 5 fw and Smarcb1 exon 6 rev. Tbp cDNA amplified with primers Tbp fw1 and Tbp rev1 was used as an internal control to measure the relative expression quantity of the target genes. Primer sequences are listed in Table 3.

**In situ hybridization**. A plasmid containing rat *Ttr* cDNA was a kind gift from Shubha Tole (Tata Institute of Fundamental Research, India)/Wei Duan (Deakin University, Australia). *Hes1* and *Hes5* mouse cDNA plasmids were kind gifts from Achim Gossler (MHH Hannover, Germany). Five μg of linearized plasmid purified with DNA Clean & Concentrator-5 (Zymo Research) were transcribed using a digoxigenin RNA labeling mix (Dig-11-UTP, Roche, #11209256910; RTP Set, Roche, #11277057001) and T3, Sp6 (Promega) or T7 RNA polymerase (Roche). Sense probes served as negative controls. Deparaffinized and rehydrated sections were incubated with 7 μg/mL proteinase K/TE buffer at RT for 15 min, refixed with 4% paraformaldehyde/PBS at RT for 20 min, briefly rinsed in water, acetylated in 0.0025% acetic anhydride/0.1 M triethanolamine for 10 min, washed in PBS for 5 min, dehydrated in increasing concentrations of ethanol, and air-dried. After a pre-hybridization step, 2 h at 55 °C in hybridization buffer, containing 50% deionized formamide, 10% dextran sulfate, 1 mg/mL yeast tRNA, 1× Denhardt's solution, 0.2 M sodium chloride, 0.01 M Tris buffer (pH 7.5), 0.05 M EDTA, 0.005 M NaH₂-PO₄×H₂0, 0.005 M Na₂HPO₄×2H₂0, slides were incubated in a humidified chamber with DIG-labeled RNA probes in hybridization buffer at 55 °C overnight. After hybridization, sections were washed with Wash solution (50% formamide, 0.15 M sodium chloride, 0.015 M sodium citrate and 0.1% Tween-20, pH 7.0) two times, 1 h each at 65 °C, followed by two washes with MABT (0.1 M maleic acid, 0.15 M NaCl, and 0.1% Tween-20), 1 h each at RT. Washed slides were treated with blocking solution (20% Normal Goat Serum in MABT), incubated with alkaline phosphatase-coupled anti-DIG antibody in blocking solution overnight, and extensively washed in MABT and TNMT (0.1 M Tris-HCl, pH 9.5, 0.1 M NaCl, 50 mM MgCl₂ and 0.1% Tween-20.) The color reaction was started by adding NBT and BCIP in TNMT buffer with 5 mM levamisole. The reaction was performed in the dark under coverslips at RT and stopped by washing in PBST. After washing and dehydrating, the sections were mounted in Kaiser's Glyceringelatine (Merck, Darmstadt, Germany). For consecutive nuclear fast red staining, the slides were

incubated in 0.1% nuclear fast red (Merck, Darmstadt, Germany) for 2 min, rinsed with water, dehydrated, and mounted in Kaiser's Glyceringelatine (Merck, Darmstadt, Germany). Images were acquired using an Aperio CS2 slide scanner (Leica Biosystems).

**Image analysis**. A cortical region above the lateral ventricle (three mutant and three littermate control E14.5 embryos were included) was selected with one border defined by the dorsomedial corner of the lateral ventricle. The width of the analyzed area along the lateral ventricle surface was 384 μm. The thickness of Sox2-, Tbr2-, and Ctip2-positive layers was calculated as an average of three measurements at 0, 192, and 384 μm. To determine the cell density within the three layers, cells expressing Sox2, Tbr2, or Ctip2 were manually counted using CellCounter plugin of the ImageJ software (https://imagej.nih.gov/ij/), National Institutes of Health, Bethesda, MD, USA). Double-positive (Sox2⁺Tbr2⁺) cells were excluded from counting. The cell density was determined as the number of single Sox2-, Tbr2-, or Ctip2-positive cells within the respective layer divided by the area of the layer. The data were expressed as the number of cells/100 μm².

**Statistical analysis**. GraphPad Prism (GraphPad software San Diego, CA) was used for graphical display. Statistical tests were performed with GraphPadPrism and Excel 2010 (Microsoft, USA). The Shapiro–Wilk test was used to test the normal distribution of data within groups. Student's *t* tests (two-tailed, unpaired) were applied to determine if datasets are significantly different. n.s. not significant ($p > 0.05$), *$p \leq 0.05$, **$p \leq 0.01$, ***$p \leq 0.001$. Measurements were taken from distinct samples. Bars display the mean with standard deviations.

**Reporting summary**. Further information on research design is available in the Nature Research Reporting Summary linked to this article.

## Data availability
The authors declare that all data supporting the findings of this study are available within the article and its supplementary information files or from the corresponding author upon reasonable request. The source data underlying Figs. 1f, g, 4b, c, and 5d, e and Supplementary Figs. 1a–e and 5a, d are provided as Source Data.

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

## Acknowledgements

We are grateful to the families for participating in this study. We thank Michaela Becker-Röck, Meike Stotz-Reimers, and Michael Wolf for excellent technical help. Thanks to Dorothea Schulte for very valuable comments on the text and figures. We thank Hermann-Josef Lüdecke for providing molecular data, and Sophie Hinreiner for excellent education of Linda Rey. This work was supported by the German Ministry of Education and Research in the frame of the E-RARE network CRANIRARE-2 to D.W. (BMBF 01GM1211B) and the Chromatin-Net consortium to D.W. (BMBF 01GM1520B). M.B.L received a stipend from the German Research Foundation (GRK 1657).

## Author contributions

A.F. designed and performed experiments, analyzed data, and wrote the paper; L.K.R. analyzed patient data and wrote the paper; M.B.L. performed experiments, analyzed data, and reviewed the manuscript text; M.H., R.P., K.S., and G.W.E.S. collected and analyzed patient data and reviewed the manuscript text; J.S. analyzed MRI scans and wrote the paper; D.W. analyzed patient data and wrote the paper; U.A.N. initiated the study, designed and performed experiments, analyzed data, and wrote the paper.

## Additional information

**Competing interests:** The authors declare no competing interests.

