## [Peer Review File · Nature Communications]

Reviewers' comments:

Reviewer #1 (Remarks to the Author):

I have read the manuscript 'Mutations in SMARCB1 and in other genes causing Coffin-Siris syndrome lead to a variety of brain midline defects' with great interest. The authors describe how they extensively characterized a mouse model with 30% reduced SMARCB1 expression which leads to several brain midline defects, and they find similar defects upon re-analysis of MRI images of CSS patients. They show clear involvement of SMARCB1 in brain development. They also propose a model why SMARCB1 haploinsufficient mice do not show similar defects. The manuscript is well written, although I find the sections on mice brain anatomy somewhat lengthy, but I do have some significant comments which should be addressed by the authors.

Major comments

- My major comment is that the authors seem to consider their mouse model a CSS model and I am not sure about this. Their model leads to about 30% expression. However, in patients specific missense mutations in the 3' end lead to CSS, whereas one specific mutation in the 5' end leads to a completely different and very specific phenotype (PMID: 29907796). Although it is possible that the mechanism leading to the phenotype is partly loss of function, it is also possible that there is a dominant negative or gain of function effect arising from the clustering missense mutations. Thus, the authors should clearly discuss the limitations of their model, and therefore of their conclusions. It is clear to me that SMARCB1 plays an important role in brain development, but to what extent this model is a mouse model of CSS remains to be seen.

- Related to this, there actually appears to be more overlap with the other SMARCB1-related ID phenotype, especially since there is increased choroid plexus tissue, with all 4 published patients showing this specific phenotype as well (PMID: 29907796). This should be discussed in much more detail, perhaps at the cost of the extensive sections on brain anatomy in mice, also together with my first comment on the chosen model. The authors argue that the absence of microcephaly in the patient group with 5' mutations would suggest that the current mouse model links to CSS, but in this group the microcephaly could be masked by the hydrocephalus, caused by the choroid plexus hyperplasia.

- Although I applaud the integration with clinical information, it is very unfortunate that only six patients are included, and in particular only 2 SMARCB1 patients, where many more have been published. Although there is significant overlap between CSS mutation groups, there is also significant difference. Importantly, microcephaly does not seem to be a feature of ARID1B mutations (see PMID 30349098), whereas it is relatively common in SMARCB1 (see eg PMID: 25168959). Throughout the manuscript, microcephaly is cited as a key feature of CSS, but this highly depends on the causative mutation, and since ARID1B is the most frequent cause of CSS I would either change it throughout the manuscript, or refer more specifically to SMARCB1-related CSS. In my view the paper would gain in quality if it would be restricted to SMARCB1-related CSS, and collect MRIs of more SMARCB1 patients. This should be feasible in a couple of months.

- The authors fail to explain the importance of using a nervous system specific partial loss-of-function mutation. It does not allow capture of other SMARCB1-related phenotypes (such as rhabdoid tumors), which is actually an important disadvantage, as absence of such features can currently not be used to argue that this model is a CSS model.

- Compensated cerebral SMARCB1 expression in SMARCB1 haploinsufficient mice is an important finding. In sup fig 5, the authors show this, but unfortunately they do not include the expression

results of their own mice in this same figure. Given that this is a key result, the authors should perform the experiment of all 4 mouse groups at the same time and include the result in one figure. In addition, the authors should provide more of a discussion about the potential mechanisms of this compensatory expression.

Minor comments

- In the abstract authors should mention that it is not a 'normal' loss-of-function mutation, but a partial loss of function mutation, or they should mention expression levels.
- Similar to microcephaly (see major comments) the majority of CSS patients does not have severe delay (page 3). This is however true for the SMARCB1-subset. Authors should restrict this statement to SMARCB1-related CSS. Since scoliosis is also a frequent feature of SMARCB1-related CSS they might consider including this in their enumeration.
- The authors could consider moving their model (p20) to the end of the discussion, since to me at least it seems all data culminates in this model.

Reviewer #2 (Remarks to the Author):

Authors created mice with a heterozygous CNS-specific loss-of-function mutation in *Smarcb1*, one of BAF complex subunits, whose germline mutations were previously found in Coffin-Siris syndrome (CSS). SMARCB1 mutations were also found in tumor predisposition syndromes. It is well known that SMARCB1 mutations in CSS are basically missense or in-frame mutations, but are various types (including truncation ones). In tumor predisposition syndromes, somatic loss of heterozygosity or second hit leads to malignant tumor formations. The very unique point in their mouse model is *Smarcb1*^{+/inv} *NesCre*^{+/-} with a heterozygous reversible *Smarcb1* disruption in neural stem/progenitor cells. Interestingly, the mice only showed brain abnormalities but no other CSS specific features. Authors claim that this is the first model mice mimicking CSS. This reviewer recognized potential important points which could contribute to the solution of enigma regarding why the constitutional *Smarcb1*^{+/1} mice never show any abnormal phenotype. Authors provided some solid evidence which may likely contribute to the brain phenotype in a detailed manner. Various pathological investigations are sound and solid, strongly supporting the midline abnormality mechanisms. This reviewer has a few but important questions/comments which may be useful to make this manuscript much clearer and understandable.

1. The most critical issue in this manuscript is the status of the mutant transcript as well as wild type transcript in the mice of *Smarcb1*^{+/inv} *NesCre*^{+/-} with a heterozygous reversible *Smarcb1* disruption in neural stem/progenitor cells. Authors showed RT-PCR study targeting a wild type allele, *Smarcb1*^{inv} allele (no inversion) and *Smarcb1*^{inv} *Cre*⁺ in forebrain tissues. More appropriate quantitative methods should be designed such as real-time quantitative PCR/Taqman, Northern blots or even Western blots to see transcript or protein status. Similar experiments should be done in the constitutional heterozygous knockout mice, too, as authors claim that such heterozygous knockout does not decrease the total amount of transcript due to the wild-type allele compensation. This reviewer thinks such a wild-type allele compensation might have occurred in the mice of *Smarcb1*^{+/inv} *NesCre*^{+/-}. This additional experiment could give us more comprehensive explanation why the constitutional heterozygous knockout did not show any phenotypes.

2. Authors showed MRI finding in the patients with a mutation in other BAF subunit genes than SMARCB1. This way to show CSS patients with different genotypes (different mutant genes) could be

misleading. So more careful consideration (or discussion) may be needed.

3. Serial pathological examination of brain parts can be done using the constitutional heterozygous knockout mice so that such mice do NOT show any abnormality in the same experimental level.

4. For quantitative PCR on *Smarcb1* $+/+$ and *Smarcb1* $+/-$ forebrain, authors designed two sets of primers on exon 1-2 and exon 5-6. The result using these primers suggests that total level of *Smarcb1* mRNA translated from two alleles was not significantly different between two models. According to the previous report by Guidi in 2001, the artificial allele disrupted by coding insertion in intron 3 had certain expression on whole mount staining of embryos for β -galactosidase activity. This reviewer thinks additional experiment using another set of primers for evaluating the amount of mRNA that contains undisrupted ex3-4 junction might help to show whether the amount of *Smarcb1* mRNA translated from wild-type allele on *Smarcb1* $+/-$ mouse is different from *Smarcb1* $+/+$ model.

Reviewer #1:

I have read the manuscript 'Mutations in SMARCB1 and in other genes causing Coffin-Siris syndrome lead to a variety of brain midline defects' with great interest. The authors describe how they extensively characterized a mouse model with 30% reduced SMARCB1 expression which leads to several brain midline defects, and they find similar defects upon re-analysis of MRI images of CSS patients. They show clear involvement of SMARCB1 in brain development. They also propose a model why SMARCB1 haploinsufficient mice do not show similar defects. The manuscript is well written, although I find the sections on mice brain anatomy somewhat lengthy, but I do have some significant comments which should be addressed by the authors.

Major comments

- My major comment is that the authors seem to consider their mouse model a CSS model and I am not sure about this. Their model leads to about 30% expression. However, in patients specific missense mutations in the 3' end lead to CSS, whereas one specific mutation in the 5' end leads to a completely different and very specific phenotype (PMID: 29907796). Although it is possible that the mechanism leading to the phenotype is partly loss of function, it is also possible that there is a dominant negative or gain of function effect arising from the clustering missense mutations. Thus, the authors should clearly discuss the limitations of their model, and therefore of their conclusions. It is clear to me that SMARCB1 plays an important role in brain development, but to what extent this model is a mouse model of CSS remains to be seen.

Answer:

The described transgenic mice harbor a heterozygous *Smarcb1* mutation in Nestin-expressing cells of the nervous system and their phenotypic features, in particular their brain defects, are similar to abnormalities found in two human developmental disorders caused by heterozygous germline *SMARCB1* mutations: CSS and another *SMARCB1*-related ID phenotype described by Kleefstra and colleagues (PMID: 29907796), which we refer to as *SMARCB1*-related ID-CPH.

To clarify these phenotypic similarities, we had compared six features of the mutant animals with these two human entities side-by-side in Table 2. We agree with this reviewer, that the frequency of one feature, bilateral choroid plexus hyperplasia, resembles more the ID-CPH patients (100% in mice and patients). The frequency of other features, however, is more consistent with *SMARCB1*-CSS, namely, microcephaly (100% in mice and patients) and growth impairment (100% in mice and patients). Since we also found an enlarged bilateral choroid plexus in one CSS individual with an *ARID1B* mutation, we suspect that bilateral choroid plexus hyperplasia might also be found in CSS, albeit at a lower frequency.

The fact that the range of brain midline defects present in the mouse model led to the discovery of corresponding and so far unrecognized abnormalities (reduction or absence of several commissures, absence or hypoplasia of the septum pellucidum, choroid plexus hyperplasia) in CSS individuals, even with mutations in other CSS-causing genes, underscores the similarity of the mouse model to this syndrome.

We have now changed several sections in the manuscript text to make it more clear that these mice show features of both, *SMARCB1*-related CSS and *SMARCB1*-related ID-CPH.

We agree with this reviewer that the consequences of human *SMARCB1* mutations on the function of the respective protein have not been demonstrated yet. This is why we write on page 16 "The effect of these mutations on the function of the *SMARCB1* protein remains unknown.". We do not suggest that missense mutations in patients lead to a loss of function, but intentionally refer to an "altered *SMARCB1* protein product" in patients (page 19).

- Related to this, there actually appears to be more overlap with the other *SMARCB1*-related ID phenotype, especially since there is increased choroid plexus tissue, with all 4 published patients showing this specific phenotype as well (PMID: 29907796). This should be discussed in much more detail, perhaps at the cost of the extensive sections on brain anatomy in mice, also together with my first comment on the chosen model. The authors argue that the absence of microcephaly in the patient group with 5' mutations would suggest that the current mouse model links to CSS, but in this group the microcephaly could be masked by the hydrocephalus, caused by the choroid plexus hyperplasia.

Answer:

Regarding the increased choroid plexus tissue, see our answer above.

In terms of the microcephaly being absent in the *SMARCB1*-related ID-CPH patients, we rely on the clinical description provided in the original article describing these patients (Diets et al., 2018), in which it is written that "features that are frequently noted in *SMARCB1*-based CSS are impaired growth, microcephaly ..., none of which are present in the individuals we describe".

- Although I applaud the integration with clinical information, it is very unfortunate that only six patients are included, and in particular only 2 *SMARCB1* patients, where many more have been published. Although there is significant overlap between CSS mutation groups, there is also significant difference. Importantly, microcephaly does not seem to be a feature of *ARID1B* mutations (see PMID 30349098), whereas it is relatively common in *SMARCB1* (see eg PMID: 25168959). Throughout the manuscript, microcephaly is cited as a key feature of CSS, but this highly depends on the causative mutation, and since *ARID1B* is the most frequent cause of CSS I would either change it throughout the manuscript, or refer more specifically to *SMARCB1*-related CSS. In my view the paper would gain in quality if would be restricted to *SMARCB1*-related CSS, and collect MRIs of more *SMARCB1* patients. This should be feasible in a couple of months.

Answer:

We appreciate this hint and clarify that microcephaly is typical for *SMARCB1*-CSS in the revised text (pages 17, 18). Given the fact that the broad variety of midline defects present in *Smarcb1* mutant mice are also found in CSS patients with mutations other than *SMARCB1*, we believe that it is clinically highly relevant to include these patients and to not restrict this manuscript to CSS patients with *SMARCB1* mutations.

As advised by this reviewer, we have collected additional MRIs from CSS patients with *SMARCB1* mutations and included them in the revised version (patients 9-11). To provide a better overview of

all MRI findings and since we had to shorten the text, the MRI data are now listed in Table 1 and the patients are described in the new Supplementary text.

- The authors fail to explain the importance of using a nervous system specific partial loss-of-function mutation. It does not allow capture of other SMARCB1-related phenotypes (such as rhabdoid tumors), which is actually an important disadvantage, as absence of such features can currently not be used to argue that this model is a CSS model.

Answer:

We have now made it more clear in the revised text that our mutation is a nervous system-specific partial loss of function one. Complete loss of function mutations predispose to rhabdoid tumors, thus we would not expect such tumors to develop in these mice. This is not a disadvantage, but exactly reflects the situation in humans, as CSS and rhabdoid tumor predisposition syndromes are not related, and this distinction is based on the different types of mutations as we discuss on pages 16-19.

- Compensated cerebral SMARCB1 expression in SMARCB1 haploinsufficient mice is an important finding. In sup fig 5, the authors show this, but unfortunately they do not include the expression results of their own mice in this same figure. Given that this is a key result, the authors should perform the experiment of all 4 mouse groups at the same time and include the result in one figure. In addition, the authors should provide more of a discussion about the potential mechanisms of this compensatory expression.

Answer:

We have now performed the quantitative RT-PCR experiments with material from all 4 mouse groups (Smrcb1+/inv NesCre+/-, Smrcb1+/inv NesCre-/-, Smrcb1+/-, Smrcb1+/+) at the same time. These data are now included in the revised Supplementary Fig. 5. In addition, we mention a potential mechanisms of this compensation on page 18.

Minor comments

- In the abstract authors should mention that it is not a 'normal' loss-of-function mutation, but a partial loss of function mutation, or they should mention expression levels.

Answer: We have now added this information to the abstract.

- Similar to microcephaly (see major comments) the majority of CSS patients does not have severe delay (page 3). This is however true for the SMARCB1-subset. Authors should restrict this statement to SMARCB1-related CSS. Since scoliosis is also a frequent feature of SMARCB1-related CSS they might consider including this in their enumeration.

Answer: We have now restricted the statement about severe delay to SMARCB1-CSS.

- The authors could consider moving their model (p20) to the end of the discussion, since to me at least it seems all data culminates in this model.

Answer: We have changed the order of the discussion subsections as recommended by this reviewer.

Reviewer #2 (Remarks to the Author):

Authors created mice with a heterozygous CNS-specific loss-of-function mutation in *Smarcb1*, one of BAF complex subunits, whose germline mutations were previously found in Coffin-Siris syndrome (CSS). SMARCB1 mutations were also found in tumor predisposition syndromes. It is well known that SMARCB1 mutations in CSS are basically missense or inframe mutations, but are various types (including truncation ones). In tumor predisposition syndromes, somatic loss of heterozygosity or second hit leads to malignant tumor formations. The very unique point in their mouse model is *Smarcb1*^{+/-} *NesCre*^{+/-} with a heterozygous reversible *Smarcb1* disruption in neural stem/progenitor cells. Interestingly, the mice only showed brain abnormalities but no other CSS specific features. Authors claim that this is the first model mice mimicking CSS. This reviewer recognized potential important points which could contribute to the solution of enigma regarding why the constitutional *Smarcb1*^{+/-} mice never show any abnormal phenotype. Authors provided some solid evidence which may likely contribute to the brain phenotype in a detailed manner. Various pathological investigations are sound and solid, strongly supporting the midline abnormality mechanisms. This reviewer has a few but important questions/comments which may be useful to make this manuscript much clearer and understandable.

1. The most critical issue in this manuscript is the status of the mutant transcript as well as wild type transcript in the mice of *Smarcb1*^{+/-} *NesCre*^{+/-} with a heterozygous reversible *Smarcb1* disruption in neural stem/progenitor cells. Authors showed RT-PCR study targeting a wild type allele, *Smarcb1* allele (no inversion) and *Smarcb1*^{inv} *Cre*⁺ in forebrain tissues. More appropriate quantitative methods should be designed such as real-time quantitative PCR/Taqman, Northern blots or even Western blots to see transcript or protein status. Similar experiments should be done in the constitutional heterozygous knockout mice, too, as authors claim that such heterozygous knockout does not decrease the total amount of transcript due to the wild-type allele compensation. This reviewer thinks such a wild-type allele compensation might have occurred in the mice of *Smarcb1*^{+/-} *NesCre*^{+/-}. This additional experiment could give us more comprehensive explanation why the constitutional heterozygous knockout did not show any phenotypes.

Answer:

The agarose gel shown in Fig. 1f that is labeled with wild type allele, *Smarcb1*^{inv} allele (no inversion) and *Smarcb1*^{inv} *Cre*⁺, is in fact not an RT-PCR, but a PCR using embryonic brain genomic DNA to demonstrate the respective genotypes and the inversion. We have in fact performed a quantitative method (quantitative RT-PCR), which is shown in Fig. 1g. Conventional heterozygous knockout mice have also been analyzed by quantitative RT-PCR (Supplementary Fig. 5d). Also in this case, we show additional genotyping results as an agarose gel in Supplementary Fig. 5a. To better distinguish the genomic DNA and cDNA analyses, we have now added titles (genomic DNA) to these figure panels.

2. Authors showed MRI finding in the patients with a mutation in other BAF subunit genes than SMARCB1. This way to show CSS patients with different genotypes (different mutant genes) could be misleading. So more careful consideration (or discussion) may be needed.

Answer:

Indeed, the microcephaly seen in mutant *Smarcb1* mice represents a typical feature of *SMARCB1*-CSS in humans, and this does not mirror all CSS cases with mutations in other genes. At the same time, we found a variety of midline defects present in *Smarcb1* mutant mice (absence of forebrain commissures, choroid plexus hyperplasia, cerebellar vermis hypoplasia) also in CSS patients with mutations other than *SMARCB1*. Thus, we believe that it is clinically highly relevant to include these patients and to not restrict this manuscript to CSS patients with *SMARCB1* mutations. To strengthen the *SMARCB1*-related data, we have now included three more CSS cases with *SMARCB1* mutations to the manuscript. In addition, we changed the gene symbols on top of the MRI scans in Fig. 2 to bold to highlight the different CSS-causing genes.

3. Serial pathological examination of brain parts can be done using the constitutional heterozygous knockout mice so that such mice do NOT show any abnormality in the same experimental level.

Answer:

We thank the reviewer for this comment and have now included serial coronal brain sections of 3 week-old constitutional heterozygous knockout mice (new Supplementary Fig. 5c and text on page 13). No midline abnormalities can be seen on these sections. Moreover, we added photographs of a 3 week-old *Smarcb1*^{+/-} and a *Smarcb1*^{+/+} littermate animal (new Supplementary Fig. 5b) to demonstrate that they do not differ in appearance unlike *Smarcb1*^{+/inv} *NesCre*^{+/-} mice, which exhibit growth impairment and microcephaly (Fig. 1a).

4. For quantitative PCR on *Smarcb1* ^{+/+} and *Smarcb1* ^{+/-} forebrain, authors designed two sets of primers on exon 1-2 and exon 5-6. The result using these primers suggests that total level of *Smarcb1* mRNA translated from two alleles was not significantly different between two models. According to the previous report by Guidi in 2001, the artificial allele disrupted by coding insertion in intron 3 had certain expression on whole mount staining of embryos for b-galactosidase activity. This reviewer thinks additional experiment using another set of primers for evaluating the amount of mRNA that contains undisrupted ex3-4 junction might help to show whether the amount of *Smarcb1* mRNA translated from wild-type allele on *Smarcb1* ^{+/-} mouse is different from *Smarcb1* ^{+/+} model.

Answer:

We have now performed the quantitative RT-PCR experiments with material from all 4 mouse groups (*Smarcb1*^{+/inv} *NesCre*^{+/-}, *Smarcb1*^{+/inv} *NesCre*^{-/-}, *Smarcb1*^{+/-}, *Smarcb1*^{+/+}) at the same time and included an additional primer pair spanning the exon3-exon4 junction as recommended by this reviewer. These data are now shown in the revised Supplementary Fig. 5d.

REVIEWERS' COMMENTS:

Reviewer #2 (Remarks to the Author):

Authors provided reasonable solutions in response to our previous concerns. They successfully showed transcripts reduction using different sets of primers and importantly they provided constitutional KO mice data with no transcript reduction.

One concern of this manuscript remains. This reviewer is not quite sure that it is reasonable to discuss with midline abnormalities of CSS by not only SMARCB1 variants but also SMARCE1 and ARID1B abnormalities in line with their unique Smarcb1 mice model showing 30% reduction of Smarcb1 transcripts. However it is still very interesting that their mice model was successful in mimicking brain abnormalities in human CSS.

Reviewer #2 (Remarks to the Author):

Authors provided reasonable solutions in response to our previous concerns. They successfully showed transcripts reduction using different sets of primers and importantly they provided constitutional KO mice data with no transcript reduction.

One concern of this manuscript remains. This reviewer is not quite sure that it is reasonable to discuss with midline abnormalities of CSS by not only SMARCB1 variants but also SMARCE1 and ARID1B abnormalities in line with their unique Smarcb1 mice model showing 30% reduction of Smarcb1 transcripts. However it is still very interesting that their mice model was successful in mimicking brain abnormalities in human CSS.

Answer:

We believe that it is highly relevant to include and discuss the MRI data of individuals with *SMARCE1* and *ARID1B* mutations for clinical-diagnostic and for scientific reasons:

a) The clinical presentation of CSS individuals is variable and there are certain genotype-phenotype correlations; for example, microcephaly is not a general CSS feature but characteristic for SMARCB1-CSS. Thus, it is highly important to know the spectrum of brain midline abnormalities that can occur in CSS with mutations in different genes. Furthermore, the case with an *ARID1B* mutation and voluminous choroid plexus in both lateral ventricles is of particular clinical relevance as choroid plexus hyperplasia has so far only been found in SMARCB1-related ID-CPH. This case demonstrates that it is essential to consider mutations in genes other than *SMARCB1* in patients with bilateral choroid plexus hyperplasia.

b) The functional specificity of BAF complexes depends on their subunit composition. It is therefore very important to learn that mutations in three BAF complex subunit genes can lead to the brain midline abnormalities described by us. This knowledge enhances our understanding of which of the various subunits are involved in the regulation of specific neurodevelopmental processes – in our case the development of brain midline structures.